# Antimicrobial Resistance Linked to Septic System Contamination in the Indiana Lake Michigan Watershed

**DOI:** 10.3390/antibiotics12030569

**Published:** 2023-03-14

**Authors:** Angad S. Sidhu, Faith N. Mikolajczyk, Jenny C. Fisher

**Affiliations:** Biology Department, Indiana University Northwest, Gary, IN 46408, USA

**Keywords:** antibiotic resistance, cephalosporins, β-lactamase genes, ESBL, septic systems, fecal contamination, Human *Bacteroides*, *E. coli*

## Abstract

Extended-spectrum β-lactamases confer resistance to a variety of β-lactam antimicrobials, and the genes for these enzymes are often found on plasmids that include additional antimicrobial resistance genes (ARG). We surveyed aquatic environments in the Indiana Lake Michigan watershed in proximity to areas with high densities of residential septic systems to determine if human fecal contamination from septic effluent correlated with the presence of antimicrobial resistance genes and phenotypically resistant bacteria. Of the 269 *E. coli* isolated from environmental samples and one septic source, 97 isolates were resistant to cefotaxime, a third-generation cephalosporin. A subset of those isolates showed phenotypic resistance to other β-lactams, fluoroquinolones, sulfonamides, and tetracyclines. Quantitative PCR was used to quantify human-associated *Bacteroides dorei* gene copies (Human *Bacteroides*) from water samples and to identify the presence of ARG harbored on plasmids from *E. coli* isolates or in environmental DNA. We found a strong correlation between the presence of ARG and human fecal concentrations, which supports our hypothesis that septic effluent is a source of ARG and resistant organisms. The observed plasmid-based resistance adds an additional level of risk, as human-associated bacteria from septic systems may expand the environmental resistome by acting as a reservoir of transmissible resistance genes.

## 1. Introduction

The β-Lactam antibiotic family is the most prescribed class of antibiotics worldwide, making up 65% of the antibiotic sales in the USA alone [1]. β-Lactams encompass a wide range of drug families, including penicillins, cephalosporins, monobactams, and carbapenems, each consisting of various classes and generations. These drugs are all characterized by their active component, the β-lactam ring, which binds to penicillin-binding proteins (PBP) and inhibits peptidoglycan synthesis, resulting in the loss of bacterial viability [2]. Recently, increased detection of β-lactam-resistant Enterobacteriaceae is threatening the efficacy of these drugs in combating bacterial diseases.

First detected in 1979, extended-spectrum β-lactamases (ESBL) are enzymes that bind to and hydrolyze the β-lactam ring, rendering a wide variety of cephalosporin classes ineffective—including cefotaxime [3]. The β-lactamase family itself consists of four classes, A–D, with class A having the most predominant ESBL genes detected globally currently, including *bla*_TEM_, *bla*_SHV_, and *bla*_CTX-M_. The *bla*_CTX-M_ family is divided into five main phylogenetic groups: CTX-M-1, CTX-M-2, CTX-M-8, CTX-M-9, and CTX-M-25, of which the first three are the most prevalent [4]. The CTX-M-1–15 variant is the most dominant across Asia, Africa, Europe, and North America, followed by CTX-M-9–14 [5]. In addition to the predominant class A genes, class C AmpC β-lactamases, encoded by the CMY-2 gene, are known to confer broad spectrum resistance to β-lactams and may be found on either plasmids or chromosomes [6].

ESBL and CMY-2 encoding genes are primarily located on transferable plasmids and can spread among bacteria via horizontal gene transfer (HGT) mechanisms [7]. The ability to spread antibiotic resistance genes via conjugation allows clinical and non-clinical environments to become reservoirs for the dissemination and proliferation of ARG in the environment, including pathogenic *E. coli* strains identified globally in surface waters [8,9,10,11]. The widespread use of antibiotics in pharmaceutical and agricultural sectors, and their resulting release into the environment, has continuously driven selection towards resistant organisms in settings such as groundwater runoff, wastewater treatment plants, and local creeks and streams [12]. Natural resistance has been previously observed in aquatic environments; however, the augmentation of the environmental resistome via human fecal pollution may serve to accelerate this process.

Waterbodies in proximity to residential areas with a high density of septic systems, such those within the Indiana Lake Michigan watershed and many other coastal or rural areas, may be impacted by septic effluent in a variety of ways [13,14]. Contamination by commensal bacteria, such as *E. coli*, has already been linked to overflows from failing septic tanks in coastal waters, well-waters, and even the Great Lakes [15,16,17]. Furthermore, effluent from onsite sanitation systems has been shown to spread antibiotic-resistant bacteria (ARB) into the environment via human fecal pollution [18]. The presence of ARB in aquatic systems presents a serious public health issue, particularly for waters with designated uses such as recreation [19], drinking water intake, and crop irrigation [20]. Therefore, more tracking is required to assess the threat from potential sources of ARB, particularly those that may disseminate resistance genes in environmental reservoirs. In this study, we examined the connection between septic system contamination and the prevalence of antibiotic resistance genes and resistant organisms in nearby aquatic environments.

## 2. Results

### 2.1. E. coli Concentrations in Water Samples

*E. coli* concentrations varied widely within waterbodies at different locations and on different sampling days (Table 1). Over two-thirds of the samples (69%) exceeded the state water quality criterion (235 MPN *E. coli*/100 mL) for bacterial ambient water quality and full-contact recreational use, and two samples exceeded the capacity of the assay, with a most probable number of *E. coli* >2419.6 per 100 mL. The Human *Bacteroides* (HB) assay detected the *Bacteroides dorei* 16S rRNA gene (DNQ or higher values) at 23 sites and was present in high concentrations (>1000 CN/100 mL) in six different samples from three waterbodies: Deer Creek, Salt Creek, and Trail Creek. Deer Creek had three positive samples, including one sample with values > 6 × 10^5^ CN/100 mL, which is nearly 10% of the value of pure septic waste (~7 × 10^6^ CN/100 mL).

*E. coli* were isolated from all sites, with 1–12 isolates obtained per sample. The percent of isolates that could grow in the presence of cefotaxime (CTX) ranged broadly (8.3–100%) and was often variable among sites within the waterbody or samples collected at different times. The number of resistant isolates tended to be higher from sites with higher densities of *E. coli* and HB copy numbers, but some sites had a low yield of resistant isolates despite significant human fecal inputs. For example, sample E00336 from Salt Creek had only 1/12 *E. coli* isolates that showed resistance to CTX, despite environmental concentrations of >900 MPN/100 mL and HB > 8000 CN/100 mL.

### 2.2. Phenotypic Antimicrobial Resistance of Isolates

A total of 269 *E. coli* were isolated from 29 samples collected from sites adjacent to areas with residential septic systems (Table 1). Isolates were obtained from Burns Ditch (*n* = 20), Coffee Creek (*n* = 35), Damon Run (*n* = 4), Deer Creek (*n* = 33), Dunes Creek (*n* = 24), Lee Creek (*n* = 2), the Little Calumet River (*n* = 9), Long Beach (sand, lake water, and outfall; *n* = 39), Salt Creek (*n* = 22), Sand Creek (*n* = 24), Smith Ditch (*n* = 3), and Trail Creek (*n* = 36). A sample from a septic system pump-out (*n* = 11 isolates), served as a reference source (Table 1). Of the 269 isolates, 97 (36.1%) showed resistance to cefotaxime (CTX). Isolates that were not resistant to CTX were not tested further. A subset of isolates (*n* = 36) was examined for resistance to additional classes of antibiotics. Of the CTX-resistant isolates, the most common type of resistance was other β-lactams, AMC (69.4%), and FOX (61.1%; Figure 1). Resistance to SXT (36.1%), CIP (30.6%), and TET (30.6%) was also observed. Two isolates showed resistance to FEP, while all others were susceptible. All isolates were susceptible to CT. Among the MDR isolates, four showed resistance to at least one drug in all five classes tested (I00600, I00643, I00652, and I00657; Table 2), and seven isolates were resistant to four classes of antibiotics. The remaining isolates were resistant to only one (*n* = 23) or two (*n* = 2) classes. The 23 single-class resistant isolates were resistant only to *β*-lactams and displayed the most common resistance phenotype AMC-FOX-CTX (*n* = 20) or AMC-CTX (*n* = 3). Several resistance phenotypes included CTX, SXT, and TE: CTX-CIP-SXT-TE (5), AMC-CTX-FOX-CIP-SXT-TE (2), CTX-FEP-CIP-SXT-TE (2), CTX-SXT (2).

Minimum inhibitory concentration (MIC) tests for CTX confirmed that all isolates were resistant to the 3rd-generation cephalosporin CTX, with MICs ranging from >6 to >48 μg/mL compared to the clinical breakpoint of 4 μg/mL. However, none of the isolates’ MICs for the 4th-generation cephalosporin FEP exceeded the breakpoint of 16 μg/mL, with many MICs < 1 μg/mL (Table 2). Isolates with the highest MICs (>32 μg/mL) and lowest MICs (6 < x < 8 μg/mL) for CTX typically showed resistance only to beta-lactams, while MDR isolates had MICs of 12–24 μg/mL.

### 2.3. Genetic Determinants of Antimicrobial Resistance in Isolates and the Environment

All CTX-resistant isolates had one or more plasmids of varying sizes, as visualized by gel electrophoresis of plasmid prep extractions. Genes associated with resistance to β-lactams and fluroquinolones were amplified from plasmids and environmental DNA from sample sites (Figure 2). The genes most frequently detected in CTX-resistant *E. coli* were CMY-2 (100%) and *bla*_CTX-M-1–15_ (96.8%). Other genes were amplified in the majority of isolates, including *bla*_CTX-M-9–14_ (77.4%), *bla*_SHV-2_ (61.3%), *bla*_KPC_ (58.1%), and *qnrS* (70.9%). The *mcr1* (colistin resistance) or *bla*_NDM_ genes were not detected in any of the isolates.

Resistance genes were less frequently detected in community DNA from water samples, which is not surprising given the relatively low abundance of *E. coli* in natural bacterial communities. CMY-2, which was found in 100% of plasmid extracts and the septic sample, was not found in any environmental DNA (Figure 2). Similarly, very few samples showed positive amplification for *bla*_CTX-1–15_, which was the second most common gene in isolates. The β-lactamase-associated genes were intermittently detected in the same waterbodies at different times or sites within the waterbody. The *bla*_SHV-2_ gene was amplified most frequently (in 55.3% of environmental DNA samples), followed by *bla*_CTX-9–14_, *bla*_KPC_ (31%), and *qnrS* (27.6%).

Table 3 shows the isolates, their sites, and samples of origin, and the genes that were amplified from each. Genes given in bold are genetic determinants of β-lactam resistance. Genes that amplified in plasmid DNA from isolates were frequently not detected in the environment from which they came, with the exception of the septic sample. Occasionally, a gene was amplified in environmental DNA but not in any of the isolates from that sample. For example, sample E00205 was positive for *bla*_CTX-1–15_ and *bla*KPC. The only concordance among the environmental and plasmid DNA was the highly human-impacted sample E00215 from Deer Creek, with five of the same genes amplified in both and one additional gene from the isolate.

### 2.4. Environmental Factors Contributing to Antimicrobial Resistance

*E. coli* concentrations across all samples did not correlate strongly with HB copy numbers (Kendall’s *τ* = 0.103, *p* = 0.44), indicating that the fecal contamination at some of the sites was not predominately due to human fecal inputs. The percentage of CTX-resistant isolates from the first round of screening negatively correlated with either *E. coli* (*τ* = −0.189, *p* = 0.85) or HB (*τ* = −0.153, *p* = 0.26) concentrations, although neither relationship was statistically significant. The only significant correlation between environmental resistance and site characteristics was the number of resistance genes detected by PCR in the environmental DNA and the HB concentration of the sample (*τ* = 0.51, *p* < 0.001). The Kendall’s *τ* value of 0.51 indicates a very strong correlation, supporting our hypothesis that septic effluent is a source of resistance elements to the environment.

Principal component analysis (PCA) of the environmental sites was performed using the percent of CTX isolates from the original sample, the presence of specific resistance genes, and *E. coli* concentrations as the variables (Figure 3). PC1 explained 32.1% of the variation among the sites, and PC2 explained 20.5% of the variation. PC1 indicated that differences in the environmental sites were mainly explained by detection of the resistance genes *bla*_CTX-1–14_, *bla*_KPC_, *bla*_SHV-2_, and *qnrS*. Variation was explained, to a lesser extent, by PC2, the *E. coli* concentrations, and the percent of initial CTX-resistant isolates. While HB was not included as variable in the PCA, both the site ordination and the overlay of HB on the input variables shows that it would explain site differences similarly to the four genes listed above. This analysis suggests that the number and type of ARG in the environment drives the differences among the sites more than general fecal pollution or number of single-resistance isolates. The number and diversity of genes detected also correlates well with the degree of human fecal contamination indicated by the HB assay.

## 3. Discussion

### 3.1. Septic Systems Threaten Environmental Water Quality in the Lake Michigan Watershed

The Lake Michigan watershed in Northwest Indiana is a mix of highly urbanized and rural environments. Unincorporated areas that lack municipal sewerage, as well as many densely populated coastal areas, rely on septic systems for the management of residential waste because of the unique landscape of the coastal dunes. In addition, homeowners may neglect obligations for regular maintenance pump-out or replacement of septic tanks that are past their intended functional age [21,22]. The local soil characteristics in the Indiana coastal zone exacerbate these issues, as over 75% of soils in regions of septic use are not suitable for proper retention of septage to ensure removal of nutrients and pathogens [23]. This creates a potential risk for human health, as septage can move quickly through the highly permeable sandy soils into groundwater and surface water, introducing active pathogens along with pharmaceuticals and personal care products (PPCPs) [13,24].

All waterbodies in our study were adjacent to residential areas with septic systems as the sole or predominant waste management system. Sample sites were located within 500 feet of known septic systems or downstream from septic areas. Therefore, we expected that sites impacted by human fecal contamination would also exhibit bacterial water quality impairment [25]. Furthermore, we hypothesized that at least some of the fecal indicator organisms isolated from environmental sources would share characteristics with those from septic systems [26]. While most samples exceeded the state water quality criterion of 235 MPN *E. coli*/100 mL (Indiana Administrative Code 327 IAC 2-1.5-8), not all sites with elevated *E. coli* concentrations also showed evidence of human fecal inputs. Conversely, not all sites with evidence of human fecal contamination exceeded the bacterial water quality criterion (e.g., Dunes Creek E00210, Trail Creek E00209; Table 1). While many non-human sources can contribute to fecal contamination [27,28], low *E. coli* concentrations due to diluted septic inputs may underestimate the health risks associated with these waters.

### 3.2. Antibiotic Resistance Genes (ARG) and Antibiotic Resistant E. coli in Septic Effluent

The presence, abundance, and proliferation of antibiotic resistant bacteria and genetic elements that confer resistance in municipal wastewater has been well established [29]. Similarly, surface waters receiving wastewater effluent typically contain more ARB and resistance genes compared to unimpacted waters [30,31]. However, less is known about the contribution of nonpoint sources, such as septic systems [32], and few studies have directly assessed the presence and diversity of resistant organisms and genetic elements in septic tanks or effluent plumes [32]. Therefore, our analysis of a septic tank sample as source material provides key information about resistant *E. coli*, their genes, and community resistance markers. CTX-resistant isolates from the septic sample were generally more phenotypically resistant and harbored more ARG (Table 3) compared to isolates from most environmental samples. A notable exception was the isolates from the Deer Creek sample, which was heavily impacted by human fecal inputs (HB > 6 × 10^5^ CN/100 mL; Table 1). The community DNA sample from the septic system was positive for five of the six antibiotic resistance genes assayed, more than any environmental samples.

Several of the isolates from the septic sample and sites with the greatest human fecal contamination showed the phenotypic and genotypic resistance to multiple classes of antibiotics. However, none of the isolates were resistant to colistin or imipenem based on disk diffusion assays, nor did any show amplification of *mcr*1 and *bla*_NDM_ genes that would confer these resistance phenotypes. Neither gene was amplified from any of the community DNA samples, including the septic sample. This is likely due to the relative rarity of these types of resistance compared to cephalosphorins.

Co-occurrence of human and agricultural fecal sources in rural areas further complicates the attribution of ARB and ARG to septic systems [33,34], as farm animals are also a significant source of resistance elements and organisms to the environment [35]. Associations between the presence or abundance of human fecal indicators and ARG and/or resistant isolates are therefore commonly used to establish these links [32,34,36]. Burch et al. found that detection of the ARG *sul*1, *tetA*, and *tetX* and the integron *intl*1 in well water was significantly correlated to the number of septic systems within the drainage area [34]. A recent report from a multi-year monitoring project of a mixed-used watershed showed that all ARG tested were detected in surface waters only when human fecal markers were also present [32]. Furthermore, ARG in all samples correlated strongly with human fecal contamination, like our findings that water samples with higher HB copy numbers tested positive to more ARG (Figure 3). Mapping of ARG hotspots revealed that septic systems were the most likely source of the inputs [32].

### 3.3. Mobile Resistance Genes in Environmental Isolates Increase Potential Health Risks

Our study took a One Health approach to monitoring a common and increasingly frequent form of AMR by assessing the crossover of β-lactam-resistant bacteria from an unregulated source of human fecal bacteria (septic systems) into the aquatic environment. We were specifically interested in *E. coli* bearing plasmids encoding ESBLs, as the CDC categorizes ESBL-producing Enterobacteriaceae a “serious threat” [37]. ESBL genes are rarely found on *E. coli* chromosomes [38] since their initial jump from the chromosome of a *Kluyvera* spp. to a conjugative plasmid [39]. Therefore, it is not surprising that CTX-resistant *E. coli* isolated from our environmental and the septic sample harbored plasmids with genes conferring this resistance (Table 3).

While most *E. coli* are non-pathogenic commensals, pathogenic ESBL-producing MDR phenotypes are common and are responsible for many nosocomial infections [40]. Such organisms have been isolated from drinking water [40,41], recreational waters [19], and surface waters [11]. One of the major concerns associated with the introduction of septic system effluent into environmental waters is that they will expand the reservoir of resistance genes [41]. ESBL genes are often associated with conjugative plasmids [7,40,41], introducing the additional risk of transfer to susceptible organisms. Studies have shown that resistance plasmids can be transmitted to human commensals from a variety of sources, including companion animals [42], exogenous sources such as contaminated food [43], or drinking water. These risk scenarios are highly relevant to the Lake Michigan watershed, as waterbodies throughout the region are used for recreational activities such as swimming and fishing. Furthermore, unincorporated areas that lack municipal sewer lines may rely on well water for their drinking water source. The plasmid-based resistance observed in our environmental isolates may expand the environmental resistome by acting as a reservoir of transmissible resistance genes.

## 4. Materials and Methods

### 4.1. Study Sites, Sample Collection

The sites sampled in this study were part of a region-wide survey of the impacts of septic systems on adjacent waterbodies, specifically how they may cause beneficial use impairment due to bacterial pollution. *E. coli* concentrations were determined for each site to assess the presence and severity of fecal pollution. The Human Bacteroides (HB) assay was used to determine if human feces were present and therefore contributed to the E. coli load. We hypothesized that sites with higher levels of human fecal pollution would tend to have a greater number and diversity of resistant *E. coli*.

Samples were collected in the late spring through fall of 2019 from 12 waterbodies, including beach sand and coastal waters of Lake Michigan, and rivers and creeks that are tributaries to the lake (Figure 4, Table 1). The sampling sites included tributary waters in residential areas with a high density of septic systems. Water samples were collected into sterile, 1L Nalgene^®^ bottles (Thermo Fisher Scientific, Waltham, MA, USA). Each bottle was rinsed three times with the water to be sampled before the final composite sample was obtained. Samples were stored on ice during transport to the lab and were processed within 6 h of collection.

### 4.2. Water Quality Assessment

Colilert Quanti-tray assays (IDEXX™, Westbrook, ME, USA) were performed following the manufacturer’s instructions to estimate the most probable number of *E. coli* concentrations for each water sample. Water samples (100 mL) were also processed via vacuum filtration onto 0.45 µm pore-size filters (Thermo Fisher Scientific, Waltham, MA, USA) and then placed onto modified membrane Thermotolerant *E. coli* (mTEC) agar (Fisher Scientific, Waltham, MA, USA). The plates were incubated for 24 h at 37 °C to select for cefotaxime-resistant *E. coli*. Colonies of *E. coli* were differentiated from other enteric or environmental bacteria by their purple color.

Water samples from each site were processed in duplicate by vacuum filtration through 0.22 µm pore-sized filters (EMD Millipore Membrane Filters™, Thermo Fisher Scientific, Waltham, MA, USA). Filters were frozen and stored at −80 °C for future DNA extraction.

### 4.3. Sample Processing and Purification of Cefotaxime-Resistant Escherichia coli Isolates

Up to 12 *E. coli* colonies per sample were randomly selected from the modified mTEC plates and streaked onto Eosin Methylene Blue (EMB) agar (Thermo Fisher Scientific, Waltham, MA, USA) to obtain pure isolates. *E. coli* growth on EMB was confirmed by the presence of metallic green or dark purple colonies. Isolates were plated on modified mTEC supplemented with 4 μg/mL cefotaxime (CTX-4). Isolates that were able to grow on CTX-4 were quadrant-streaked as a final purification step to produce single colonies of pure, resistant isolates. Isolates were then stored in glycerol at −80 °C.

### 4.4. Minimum Inhibitory Concentration and Multi-Drug Resistance Tests

E-test strips were used to determine the minimum inhibitory concentration (MIC) of cefotaxime and cefepime for a subset of 36 isolates. Isolate densities were standardized in liquid medium using a 0.5 McFarland Standard and spread onto Mueller Hinton II agar (Fisher Scientific, Waltham, MA, USA) using the tri-swab method to ensure proper coverage. Cefotaxime and cefepime E-test strips (Liofilchem™ MTS™, Thermo Fisher Scientific, Waltham, MA, USA), with concentrations from 0.016 to 256 μg/mL, were placed in the middle of the streaked plates using sterile forceps. The plates were then incubated overnight at 37 °C. The concentration at which growth was inhibited was compared to the Clinical and Laboratory Standards Institute (CLSI) guidelines to classify each isolate as either resistant or susceptible.

Disc diffusion tests were performed to identify sensitivities to 8 other types of antibiotics with ranging clinical applications: Colistin (CT) (10 µg/mL), Sulfamethoxazole/Trimethoprim (SXT) (25 µg/mL), Tetracycline (TE) (30 µg/mL), Ciprofloxacin (CIP) (5 µg/mL), Cefoxitin (FOX) (30 µg/mL), Amoxicillin/Clavulanic Acid (AMC) (30 µg/mL), Imipenem (IPM) (10 µg/mL), and Cefepime (FEP) (30 µg/mL) (Hardy Disks™, Thermo Fisher Scientific, Waltham, MA, USA). Isolates were standardized and spread onto Mueller Hinton II agar plates with the same procedure used for the MIC E-tests. Four antibiotic discs were evenly spaced apart from each other on the agar plates using sterile forceps. The plates were then incubated overnight at 37 °C, and the diameter of the zone of inhibition was measured (in mm). Results were compared to the Clinical and Laboratory Standards Institute (CLSI) guidelines to classify each isolate as susceptible, intermediate, or resistant [44].

### 4.5. Environmental DNA and Plasmid Extractions

Frozen filters from environmental waters were crushed with a sterile spatula, and DNA was extracted from the filter pieces with a Powerlyzer Powersoil extraction kit (Qiagen^®^, MD, USA). The manufacturer’s protocol was modified slightly by adding the class beads and lysis reagent to the filter tube. Extracted DNA was quantified on a NanoDrop2000™ (Thermo Fisher, MA, USA) and stored at −20 °C until analysis.

Plasmid extractions were performed to identify antibiotic resistance genes (ARG) that could be mobilized in the environment. LB Broth (Fisher Scientific, Waltham, MA, USA) supplemented with both cefotaxime and yeast extract (19 g/L) was used to grow 100 mL isolate cultures for 16 h at 37 °C in a shaking incubator set to 200 rpm. The ZymoPureII™ Plasmid Midiprep kit (Zymo Research) was used with the vacuum filtration protocol for the extraction of plasmids. Plasmid DNA (30 µL) was eluted, and the purity and concentration of DNA samples were quantified using the NanoDrop2000. Extracted plasmids were stored at −20 °C. Gel electrophoresis was performed to estimate the number and approximate size of plasmids from each isolate.

### 4.6. qPCR Assays for Antibiotic Resistance Genes and Human Fecal Marker Genes

Gene amplification was performed via qPCR to detect the presence of genetic resistance determinants. Plasmids extracted from CTX-resistant isolates and DNA from the environmental water samples were tested for *bla*_CTX-M-1–15_, *bla*_CTXM-9–14_, *bla*_SHV2_, *bla*_KPC_, *bla*_NDM_, CMY-2, *qnrS*, and *mcr*1 using primer sets and cycling conditions previously described (Table 4; [45,46,47,48,49]). Final melt curve analysis and gel electrophoresis were performed to confirm that amplicon characteristics that matched those of the positive controls.

Human fecal contamination in environmental samples was quantified using the EPA Human Bacteroides HF183/BacR287 TaqMan^®^ (Thermo Fisher Scientific, Waltham, MA, USA) probe qPCR assay (HB Assay; Table 4) originally described by Green et al. [50]. All DNA samples were analyzed in duplicate, and all qPCR runs included a standard curve, technical replicates, sample spike qPCR, no template controls, and an internal standard.

### 4.7. Data Analysis

All data analyses were conducted in R [51]. Non-normality of data distribution was determined using the Shapiro–Wilk normality test in the stats package. Non-parametric correlations were performed using the Kendall method in cor.test. Principal Component Analysis (PCA) was conducted with FactoMineR [52].

## 5. Conclusions

This study investigated the impacts of septic systems on waterbodies within the southern Lake Michigan watershed with respect to bacterial contamination and, more specifically, the introduction of antibiotic-resistant bacteria into the natural environment. Through comparison of the genotypic and phenotypic traits of *E. coli* isolates from the environment and a septic source, quantification of human fecal contamination, and detection of ARG in the environment, we saw a strong correlation between the diversity of resistance potential and the impact of effluent from nearby septic systems. Our results are somewhat limited by the simple detection versus the quantification of ARG from environmental DNA. Future studies could further explore the relative abundance of ARG in each environment compared to the total bacterial abundance to better estimate the proportion of resistant organisms and provide a better correlation with the HB human fecal markers. Despite these data limitations, our study does provide further support for the observation [32] that septic systems can contribute to the expansion of the aquatic resistome.

## Figures and Tables

**Figure 1 antibiotics-12-00569-f001:**
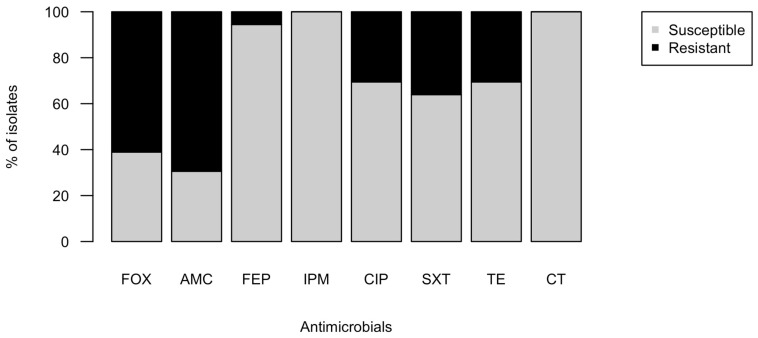
Proportion of environmental isolates resistant to common antibiotics. A total of 269 purple colonies were picked from modified mTEC plates of water samples and assayed for resistance to cefotaxime (CTX). A subset of the 97 CTXr isolates (*n* = 36) was further tested for resistance to the antibiotics FOX, AMC, FEP, IPM, CIP, SXT, TE, and CT using disk diffusion assays. Resistant and susceptible classifications were determined by the diameters of the zones of inhibition based on CSLI standards.

**Figure 2 antibiotics-12-00569-f002:**
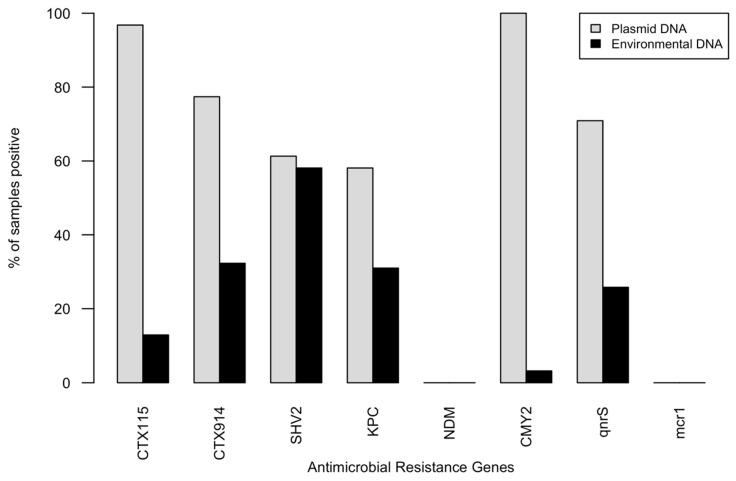
Percent of isolate plasmid DNA samples (*n* = 31) and environmental DNA samples, (*n* = 29), that had positive amplifications for genetic determinants of antimicrobial resistance.

**Figure 3 antibiotics-12-00569-f003:**
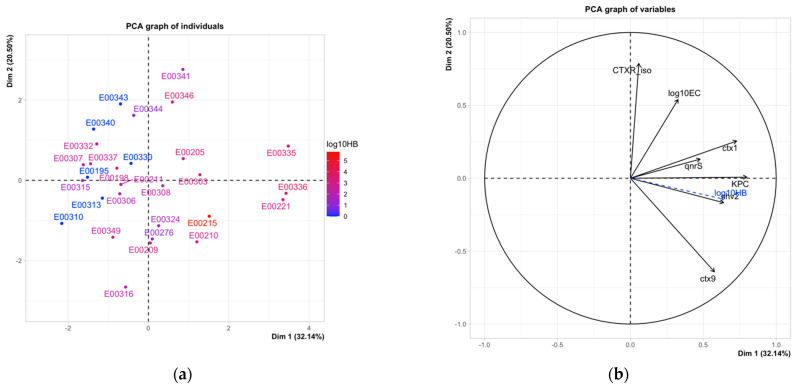
Principal component analysis of environmental sites using the input variables using percent of CTX isolates, the presence of specific resistance genes, and *E. coli* concentrations (log10EC). (**a**) Ordination of sites by the two principal component axes. Sites are shaded by the log10 concentration of HB. (**b**) Impact of variables on sample ordination. Variables shown in black were used in the analysis; log10 HB (in blue) was supplementary.

**Figure 4 antibiotics-12-00569-f004:**
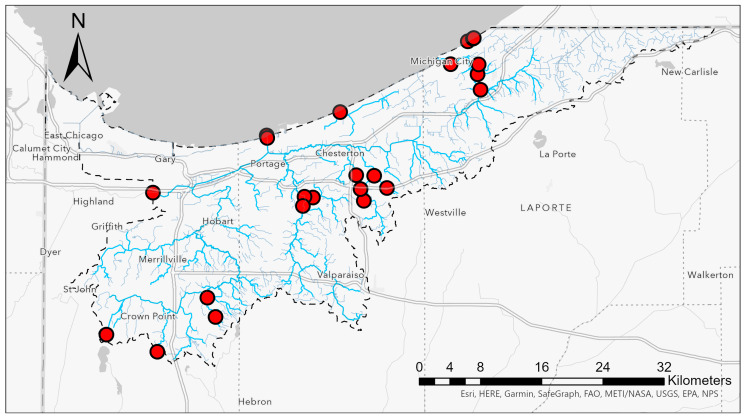
Water samples were collected from lake water and tributary creeks and streams across the Lake Michigan watershed in Indiana. Collection sites are indicated by red dots. The watershed boundary is indicated by a dashed black line.

**Table 1 antibiotics-12-00569-t001:** Most probable *E. coli* concentrations, percent CTX-resistant *E. coli*, and Human *Bacteroides* gene copy number concentrations for sampled sites.

Site Name	Sample ID	*E. coli*	% CTX^R^	HB
		(MPN/100 mL)	(of N Tested ^a^)	(CN/100 mL)
Burns Ditch	E00307	139.6	40.0 (10)	102
	E00308	113.7	30.0 (10)	219
Coffee Creek	E00324	**1020 ^b^**	8.3 (12)	40
	E00340	**980.4**	54.5 (11)	0
	E00341	**980.4**	100.0 (12)	58
Damon Run	E00337	**770.1**	50.0 (4)	387
Deer Creek	E00215	**>2419.6 ^c^**	11.1 (9)	6.14 × 10^5^
	E00303	**613.1**	12.5 (8)	1416
	E00306	**270**	25.0 (4)	78
	E00346	**344.8**	100.0 (12)	2528
Dunes Creek	E00210	193.6	8.3 (12)	857
	E00211	**307.6**	16.7 (12)	159
Lee Creek	E00330	**>2419.6**	100.0 (2)	0
Little Calumet	E00349	103.9	11.1 (9)	221
Long Beach	E00195	**450**	8.3 (12)	0
	E00198	**399**	9.1 (11)	289
	E00310	2	66.7 (3) ^d^	0
	E00313	20	33.3 (3)	0
	E00315	145	25.0 (8)	29
	E00316	1	100.0 (2) ^d^	102
Salt Creek	E00335	**816.4**	63.6 (11)	3980
	E00336	**920.8**	9.1 (11)	8467
Sand Creek	E00343	**387.3**	75.0 (12)	0
	E00344	**2419.6**	66.7 (12)	DNQ ^e^
Smith Ditch	E00332	**>2419.6**	27.3 (3)	193
Trail Creek	E00205	**285.6**	100 (1)	1571
	E00209	**182.8**	9.1 (11)	839
	E00221	**488.4**	8.3 (12)	554
	E00276	**272.3**	8.3 (12)	DNQ
Septic reference	P00041	1.46 × 10^6^	72.7 (11)	7.06 × 10^6^

^a^ Total number of *E. coli* colonies picked and grown from environmental sample plate. ^b^ Values in bold exceed the ambient water quality criterion of 235 MPN *E. coli*/100 mL. ^c^ Sample concentration exceeded maximum MPN of the Colilert^®^ assay. ^d^ Colonies were picked from replicate plates of the same sample. ^e^ DNQ: Gene was detected but the value was below the limit of quantification for the assay.

**Table 2 antibiotics-12-00569-t002:** Resistance phenotypes and cephalosporin MICs for CTX^r^ isolates.

Isolate ID	Resistance Phenotype	CTX MIC	FEP MIC
I00646	AMC-CTX	48 < x < 64	0.5 < x < 0.75
I00729	AMC-CTX	8 < x < 16	0.5 < x < 1
I00735	AMC-CTX	8 < x < 16	0.25 < x < 0.5
I00600	AMC-CTX-FOX-CIP-SXT-TE	12 < x < 16	1.5 < x < 2
I00643	AMC-CTX-FOX-CIP-SXT-TE	16 < x < 24	1.5 < x < 2
I00644	AMC-FOX-CTX	16 < x < 24	0.125 < x < 0.19
I00645	AMC-FOX-CTX	16 < x < 32	0.25 < x < 0.5
I00647	AMC-FOX-CTX	12 < x < 16	0.38 < x < 0.5
I00648	AMC-FOX-CTX	32 < x < 48	0.75 < x < 1
I00649	AMC-FOX-CTX	16 < x < 24	0.5 < x < 0.75
I00650	AMC-FOX-CTX	12 < x < 16	0.75 < x < 1
I00651	AMC-FOX-CTX	16 < x < 32	0.25 < x < 0.5
I00730	AMC-FOX-CTX	6 < x < 8	0.25 < x < 0.38
I00731	AMC-FOX-CTX	8 < x < 16	0.5 < x < 1
I00732	AMC-FOX-CTX	16 < x < 32	1 < x < 2
I00733	AMC-FOX-CTX	8 < x < 16	0.25 < x < 0.5
I00734	AMC-FOX-CTX	8 < x < 16	0.5 < x < 1
I00736	AMC-FOX-CTX	8 < x < 16	0.125 < x < 0.25
I00737	AMC-FOX-CTX	32 < x < 48	1 < x < 1.5
I00738	AMC-FOX-CTX	6 < x < 8	0.38 < x < 0.5
I00739	AMC-FOX-CTX	6 < x < 8	0.125 < x < 0.19
I00740	AMC-FOX-CTX	6 < x < 8	0.38 < x < 0.5
I00652	CTX-FEP-CIP-SXT-TE	24 < x < 32	1.5 < x < 2
I00657	CTX-FEP-CIP-SXT-TE	12 < x < 16	4 < x < 6
I00653	CTX-CIP-SXT-TE	16 < x < 24	2 < x < 3
I00654	CTX-CIP-SXT-TE	32 < x < 48	4 < x < 6
I00655	CTX-CIP-SXT-TE	16 < x < 24	2 < x < 3
I00656	CTX-CIP-SXT-TE	24 < x < 32	2 < x < 3
I00659	CTX-CIP-SXT-TE	24 < x < 32	2 < x < 3
I00613	CTX-SXT	8 < x < 12	0.75 < x < 1
I00614	CTX-SXT	8 < x < 12	1 < x < 1.5

**Table 3 antibiotics-12-00569-t003:** Comparison of AMR genes amplified in *E. coli* isolates and their samples of origin.

Sample ID	Site	Isolates	*CTX* 1–15	*CTX* 9–14	*SHV*-2	*CMY*-2	*NDM*	*KPC*	*qnrS*	*mcr*-1
E00205	Trail Creek									
		I00600								
E00209	Trail Creek									
		I00613								
E00276	Trail Creek									
		I00643								
E00215	Deer Creek									
		I00614								
E00303	Deer Creek									
		I00652								
E00306	Deer Creek									
		I00653								
E00346	Deer Creek									
		I00729								
		I00730								
		I00731								
		I00732								
		I00733								
		I00734								
		I00735								
		I00736								
		I00737								
		I00738								
		I00739								
		I00740								
E00307	Burns Ditch									
		I00654								
		I00655								
		I00656								
		I00657								
E00308	Burns Ditch									
		I00659								
P00041	Septic									
		I00644								
		I00645								
		I00646								
		I00647								
		I00648								
		I00649								
		I00650								
		I00651								

**Table 4 antibiotics-12-00569-t004:** Primer and probe sequences and cycling conditions for qPCR assays.

Assay	Primer/ Probe	Sequence	Cycles	Ref.
*CTX* 1–15	*fwd*	CGCAAATACTTTATCGTGCTGAT	95 °C for 3 min, 40 cycles of 95 °C for 5 s, 57 °C for 30 s, and 95 °C for 60 s. Final elongation at 72 °C for 7 min	[48]
	*rev*	GATTCGGTTCGCTTTCACTTT		
*CTX* 9–14	*fwd*	GCTCATCGATACCGCAGATAAT	95 °C for 3 min, 40 cycles of 95 °C for 5 s, 57 °C for 30 s, and 95 °C for 60 s. Final elongation at 72 °C for 7 min	[48]
	*rev*	CCGCCATAACTTTACTGGTACT		
*SHV*-2	*fwd*	CTGGAGCGAAAGATCCACTATC	95 °C for 3 min, 40 cycles of 95 °C for 5 s, 57 °C for 30 s, and 95 °C for 60 s. Final elongation at 72 °C for 7 min	[49]
	*rev*	CGCTGTTATCGCTCATGGTAA		
*CMY*-2	*fwd*	AGGGAAGCCCGTACACGTT	95 °C for 10 min, 40 cycles of 95 °C for 10 s, 52 °C for 30 s, and 79 °C for 17 s	[45]
	*rev*	GCTGGATTTCACGCCATAGG		
*NDM*	*fwd*	GATTGCGACTTATGCCAATG	95 °C for 3 min, 40 cycles of 95 °C for 30 s and 60 °C for 60 s	[46]
	*rev*	TCGATCCCAACGGTGATATT		
*KPC*	*fwd*	CAGCTCATTCAAGGGCTTTC	95 °C for 3 min, 40 cycles of 95 °C for 30 s and 60 °C for 45 s	[46]
	*rev*	GGCGGCGTTATCACTGTATT		
*qnrS*	*fwd*	GTGAGTAATCGTATGTACTTTTG	95 °C for 3 min, 40 cycles of 95 °C for 45 s, 52 °C for 45 s, and 72 °C for 60 s. Final elongation at 72 °C for 10 min	[47]
	*rev*	AAACACCTCGACTTAAGTCT		
*mcr*-1	*fwd*	TCCAAAATGCCCTACAGACC	94 °C for 4 min, 40 cycles of 94 °C for 5 s, 59 °C for 15. Final elongation at 72 °C for 5 min	[48]
	*rev*	GCCACCACAGGCAGTAAAAT		
*HF*183	*fwd*	ATCATGAGTTCACATGTCCG	95 °C for 10 min, 40 cycles of 95 °C for 15 s, 60 °C for 1 min	[50]
*HB*287*R*	*rev*	CTTCCTCTCAGAACCCCTATCC		
*BacP*234	*probe*	FAM-CTAATGGAACGCATCCC-MGB		

## Data Availability

Data are contained within the article.

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
