# Peer review of "Antimicrobial Resistance Linked to Septic System Contamination in the Indiana Lake Michigan Watershed"

_antibiotics, 2023, doi:10.3390/antibiotics12030569_

Round 1

Reviewer 1 Report

The article Antimicrobial Resistance Linked to Septic System Pollution in 2 the Indiana Lake Michigan Watershed the authors examined the connection between septic system pollution and the prevalence 67 of antibiotic resistance genes and resistant organisms in aquatic environments. The article tackles the major issue of antimicrobial resistance in the aquatic environment and provides a view on several contributing factors. The research uses well established techniques with control measures present.

The article is based on sampling that was completed in 2019. As this was before the Covid-19 pandemic the results attained are not accurate enough to relate to the current levels of AMR genes in these environments. The sudden rise in circulating AMR genes is thought to be due to the increase in prophylactic antibiotic use during the pandemic as primary health care workers tried to control secondary bacterial infections to the Covid-19 outbreak. 

The article is mostly well written and clear however, I could not find a clear conclusion section which should be addressed. 

I would also like to see the PCR primer codes and any reference to papers to support these.

Further, have you considered a comparison of  resistance genes to the 16s gene, to give a bacterial load to resistance gene comparison, as the sampling size was very small this may change some of the outcomes.

Author Response

I could not find a clear conclusion section which should be addressed. 

We have added a conclusion section at the end of the paper.

I would also like to see the PCR primer codes and any reference to papers to support these.

We have added a table in the methods section with the primer sequences for all qPCR reactions. The references for the primers were cited in the methods section text, but now they can easily be found along with the sequences and the cycling conditions for each reaction in the table.

Further, have you considered a comparison of  resistance genes to the 16s gene, to give a bacterial load to resistance gene comparison, as the sampling size was very small this may change some of the outcomes.

The comparison of the copy number of resistance genes and 16S rRNA genes is an excellent suggestion. At this point, we have only used qPCR for detection, not quantification. We are unable to perform such an analysis on short notice, but will include in future work with this dataset.

Reviewer 2 Report

Microorganisms must always be written in italics. I think it is better (maybe) write "contamination" than "pollution". 

Line 78: It must be written "sand", not "sands".

Figure 2: With the new CLSI criteria, there is no intermediate category, but increased exposure/dosage. Better change that.

Figure 4: It is difficult to understand what the graph wants to explain

However, I like the rest of the figures, a lot, especially the map.

Author Response

Microorganisms must always be written in italics. I think it is better (maybe) write "contamination" than "pollution". 

 We have corrected all organism names to be in italics and changed pollution to contamination.

Line 78: It must be written "sand", not "sands".

 We have corrected to sand.

Figure 2: With the new CLSI criteria, there is no intermediate category, but increased exposure/dosage. Better change that.

 We conducted these analyses based on former CLSI standards that did include intermediate, but agree that with the current standards it makes sense to only include R/S. We have changed figure 2 to reflect that.

Figure 4: It is difficult to understand what the graph wants to explain

We have added a more detailed explanation of Figure 4 in the text.

However, I like the rest of the figures, a lot, especially the map.

Thank you! We like the map very much too.

Reviewer 3 Report

This article aims to examine the connection between septic system pollution and the prevalence of antibiotic resistance genes and resistant organisms in nearby aquatic environments.

Despite this important observation, the authors must address some areas of concern.

Areas of concern:

Make sure E.coli is italicized throughout the manuscript

Abstract:

This section lacks sample collection and methodology.

Lines 14-15: This sentence is incomplete.

Methodology

Line 299: replace ‘’per’’ with ‘’following’’

Introduction

Line 54: ‘’of’’ is missing between ‘’ augmentation’’ and ‘’the’’ in the following phrase: however, the augmentation the environmental resistome

Results

Lines 71-79: take this paragraph to the methodology section. Figure 1 should also be moved to the methodology section.

Equally, move Table 1 where Figure 1 was. Table 2 should come up immediately after Figure 2

Line 142: Write and read Table 2 instead of Table 1.

Move Figure 3 to where Table 2 was previously.

Discussions

Why are the sub-headings under this section not numbered? Either you number the sub-headings or you remove the sub-headings.

The authors did not explain why mcr1 (colistin resistance) or blaNDM genes were not detected in any of the isolates.

Conclusion

A formal section for the conclusion is missing in this paper. This section should some limitations of the study that would call for other studies to elucidate some unexplained result data.

Author Response

Make sure E.coli is italicized throughout the manuscript

We have corrected all organism names to be in italics.

Abstract:

This section lacks sample collection and methodology.

We did not include this information in the abstract to meet the word limit. We felt that the results were more important than the details of sampling.

Lines 14-15: This sentence is incomplete.

We have corrected this sentence to be complete.

Methodology

Line 299: replace ‘’per’’ with ‘’following’’

We have corrected this

Introduction

Line 54: ‘’of’’ is missing between ‘’ augmentation’’ and ‘’the’’ in the following phrase: however, the augmentation the environmental resistome

We have corrected this

Results

Lines 71-79: take this paragraph to the methodology section. Figure 1 should also be moved to the methodology section.

Equally, move Table 1 where Figure 1 was. Table 2 should come up immediately after Figure 2

Line 142: Write and read Table 2 instead of Table 1.

Move Figure 3 to where Table 2 was previously.

We have rearranged the text, figures, and tables to shift the sampling information to the methods and re-ordered the numbering in the results as needed.

Discussions

Why are the sub-headings under this section not numbered? Either you number the sub-headings or you remove the sub-headings.

For some reason, the template we were given did not include numbers for the discussion. This was confusing to us as well, but we followed the example shown. We have added numbers to the subheadings. The copy editors can remove the numbering if it is not appropriate.

The authors did not explain why mcr1 (colistin resistance) or blaNDM genes were not detected in any of the isolates.

We have added our best explanation as to why these genes were not found (specifically that this type of resistance is rarer than the other types tested. We know that the reaction works as the positive controls were positive)

Conclusion

A formal section for the conclusion is missing in this paper. This section should some limitations of the study that would call for other studies to elucidate some unexplained result data.

We have added a conclusion section and provided more detail here in the discussion sections about some of our unexplained result data and potential future analyses we would like to pursue.

Round 2

Reviewer 1 Report

Thank you for your revisions however, in your conclusions chapter that you added you state that "Through comparison of genotypic and phenotypic traits of E. coli from the environment 383 and septic systems, and quantification of human fecal contamination". This statement is not backed by your results section. The results show if the resistance gene is present and not the quantity therefore, for this small sampling size an appropriate comparison of resistance gene presence cannot be made. 

Author Response

We believe the reviewer may have misread our results and interpretation-- we did indeed measure phenotypic (by disk diffusion assays) and genotypic (through amplification of resistance genes on plasmids from individual isolates) characteristics of isolates. The isolates from more highly contaminated environments shared more characteristics with those from the septic sample than the less impacted areas. We also quantified human pollution (using the HB) assay to give copy numbers of the human-associated fecal marker gene. The environmental ARG were not quantified, but we also do not make that claim in the conclusions, and specifically state that this was a limitation of the study. We have adjusted a few statements in the conclusion to better clarify this interpretation. 

This study investigated the impacts of septic systems on waterbodies within the southern Lake Michigan watershed with respect to bacterial contamination and more specifically, the introduction of antibiotic-resistant bacteria into the natural environment. Through comparison of genotypic and phenotypic traits of E. coli isolates from the environment and a septic source, and quantification of human fecal contamination, and detection of ARG in the environment we saw a strong correlation between the diversity of resistance potential and the impact of effluent from nearby septic systems. Our results are somewhat limited by the simple detection versus the quantification of ARG from environmental DNA. Future studies could further explore the relative abundance of ARG in each environment compared to the total bacterial abundance to better estimate the proportion of resistant organisms and provide a better correlation with the HB human fecal markers. Despite these data limitations, our study does provide further support for the observation [32] that septic systems can contribute to the expansion of the aquatic resistome.

Round 3

Reviewer 1 Report

Thank you for all your corrections. This paper although exhibiting limitations of a small data set (some n=2) does add weight to the current understanding of AMR entering the aquatic environment. 

There are some typos and structural issues such as splitting the conclusion through a table to be adjusted. 

Author Response

Thank you for the feedback. I would hope that the editorial staff / copy editors would fix any final formatting issues. I've never submitted to a journal where I was expected to be the publisher, so I hope that they will handle any small details like that.